# The Potential of Monitoring Carbon Dioxide Emission in a Geostationary View with the GIIRS Meteorological Hyperspectral Infrared Sounder

**Qi Zhang** [1,*] **, William Smith, Sr.** [1] **and Min Shao** [2]

1   Space Science and Engineering Center, University of Wisconsin-Madison, 1225 W. Dayton St.,
    Madison, WI 53706, USA
2   School of Environment, Nanjing Normal University, Nanjing 210023, China
*   Correspondence: qzhang487@wisc.edu

**Abstract:** With the help of various polar-orbiting environment observing platforms, the atmospheric concentration of carbon dioxide ($CO_2$) has been well established on a global scale. However, the spatial and temporal pattern of the $CO_2$ emission and its flux dependence on daily human activity processes are not yet well understood. One of the limiting factors could be attributed to the low revisit time frequency of the polar orbiting satellites. With high revisiting frequency and $CO_2$-sensitive spectrum, the Geostationary Interferometric Infrared Sounder (GIIRS) onboard the Chinese FY-4A and FY-4B satellites have the potential to measure the $CO_2$ concentration at a higher temporal frequency than polar-orbiting satellites. To provide a prototypical demonstration on the $CO_2$ monitoring capability using GIIRS observations, a hybrid-3D variational data assimilation system is established in this research and a one-month-long experiment is conducted. The evaluations against the Goddard Earth Observing System version 5 (GEOS-5) analysis field and Orbiting Carbon Observatory -2/-3 (OCO-2/-3) $CO_2$ retrieval products reveal that assimilating GIIRS observations can reduce the first guess's $CO_2$ concentration mean bias and standard deviation, especially over the lower troposphere (975–750 hPa) and improve the diurnal variation of near surface $CO_2$ concentration.

**Keywords:** geostationary; hyperspectral; infrared; $CO_2$; monitoring; assimilation



## 1. Introduction

The global-scale atmospheric carbon dioxide ($CO_2$) concentration has been monitored through ground observation networks, e.g., the World Data Centre for Greenhouse Gases (WDCGG) [1] and Total Carbon Column Observing Network (TCCON) [2], for a long time, which provides a fundamental contribution on revealing the fundamental climatological impact from Green House Gases (GHG). However, the spatial and temporal uncertainties from underlying processes, e.g., the diurnal flux variation caused by human activities and international transportation, are not yet well quantified [3,4].

Observation from space holds immense potential to settle the pre-existing ground-based network's spatial coverage problem. The Greenhouse gas Observing Satellite (GOSAT) and Greenhouse gas Observing Satellite-2 (GOSAT-2), launched in 2009 and 2018, can monitor the atmospheric $CO_2$ concentration from the Thermal and Near infrared Sensor for carbon Observations-Fourier Transform Spectrometer (TANSO-FTS) with a 10.5 km horizontal resolution [5]. The OCO-2 and OCO-3, launched in 2014 and 2019, monitors the $CO_2$ concentration at high spatial resolution (1.29 km cross-track, 2.25 km along-track) with the grating spectrometer [6,7]. The meteorological hyperspectral infrared sounders, designed for high-vertical-resolution temperature and water vapor profiles, are also capable of measuring variations in carbon trace gasses such as $CO_2$, e.g., Atmospheric Infrared Sounder (AIRS) [8–11], Infrared Atmospheric Sounding Interferometer (IASI) [12–14], Cross-track Infrared Sounder (CrIS) [15–17], and Infrared Fourier Spectrometer-2 (IKFS-2) [18].

Recently, the need for $CO_2$ concentration observation with a much higher spatiotemporal resolution has led us to the Geostationary Carbon Cycle Observatory (GeoCarb), which will deliver $CO_2$ concentration observation on (at least) a daily basis at fine (10 km) spatial scales over the Americas [19]. Meanwhile, such demand also gives rise to investigations on exploiting the meteorological geostationary hyperspectral infrared sounder's $CO_2$ concentration monitoring capability, e.g., GIIRS [20], Meteosat Third Generation–InfraRed Sounder (MTG-IRS) [21], and Geostationary Extended Observations (GeoXO) Hyperspectral InfraRed Sounder [22]. In addition to $CO_2$ monitoring, observations from geostationary hyperspectral infrared sounders have been used in estimating other trace gas emissions, e.g., ammonia [23,24].

This paper demonstrates the geostationary hyperspectral infrared sounder's capability to provide $CO_2$ monitoring products with reasonable spatial, but much higher temporal, resolution by generating a one-month-long (from 1 August 2022 to 1 September 2022) $CO_2$ profile dataset at 16 km horizontal and 3 h temporal resolution. Section 2 describes the basics of the $CO_2$ data assimilation system and the usage of GIIRS observation. Section 3 presents results and relevant error statistics. Section 4 is the summary.

## 2. Materials and Methods

The GIIRS onboard the Chinese FY-4A and FY-4B satellites (data available at https://satellite.nsmc.org.cn/portalsite/ (accessed on 31 December 2022)) can measure the $CO_2$ concentration with scanning frequencies of ten times per day (at 0, 2, 4, 6, 8, 10, 12, 14, 20, and 22 h UTC) and a spectrum sampling rate of 0.625 cm$^{-1}$. To avoid interference from other trace gases and retain $CO_2$ information content, we selected 22 spectral channels with $CO_2$ (other trace gases) absorbance higher (lower) than 0.6 (0.3) within the long-wave IR band (700–1130 cm$^{-1}$) (Figure 1a). The absorbance dataset in this study comes from the HIRTRAN 2020 dataset (available at https://hitran.org (accessed on 31 December 2022)). For further validation of this channel selection scheme, we calculated the CO2 Jacobian for each selected channel using the European Organization for the Exploitation of Meteorological Satellites (EUMETSAT) 60-level sample profile dataset from the Monitoring Atmospheric Composition and Climate (MACC) project (available at https://nwp-saf.eumetsat.int/site/download/profile_datasets/60l_macc.dat.tar.bz2 (accessed on 31 December 2022)). The Jacobian peak pressure level (Figure 2) indicates that the $CO_2$ information content from the selected spectrum mostly stays below 100 hPa, and the last six spectral channels (central wavenumber larger than 751.250 cm$^{-1}$) can modify the low-troposphere $CO_2$ concentration. Even though there are extra absorption channels located within the GIIRS mid-wave IR channel (1650–2250 cm$^{-1}$) (Figure 1b), due to their low sensitivity (compared to selected long-wave IR spectrum) to $CO_2$, these spectral observations are not used in this study. The channel selection differences among GIIRS, AIRS, and IASI [8,12] can be found in Figure 3. As can be seen, GIIRS channel selection scheme adds five channels whose wavenumber is higher than 750 cm$^{-1}$ (751.250 cm$^{-1}$, 758.750 cm$^{-1}$, 791.250 cm$^{-1}$, 791.875 cm$^{-1}$, 792.500 cm$^{-1}$) to increase its low troposphere $CO_2$ inversion capability, since the weighting function peak level of this spectrum stays below 800 hPa (Figure 2).

Generating $CO_2$ concentration profiles using a variational Data Assimilation (DA) method can ensure that the $CO_2$ added value and meteorological fields are consistent with each other [25]. Particularly, the atmospheric temperature produces the most significant uncertainty to $CO_2$ inversion for two reasons: (1) $CO_2$-sensitive spectral observations are highly related to atmospheric temperature, and (2) the channel selection scheme renders the water vapor and other trace-gas side effects to a low degree on $CO_2$ profile retrieval. In this study, the priori $CO_2$ and meteorological background is derived from the NASA (National Aeronautics and Space Administration) GEOS-5 48 h lead-time forecast (data available at https://portal.nccs.nasa.gov/datashare/gmao/geos-fp/forecast (accessed on 31 December 2022)) [26]. Taking advantage of the hybrid 3-Dimensional Variational (hybrid 3DVar) assimilation scheme [27], the DA system adjusts the background using 22 GIIRS

spectral radiance observations. The L-BFGS-B method [28] is chosen to minimize the cost function:

$$J_{(x)} = (x - x_b)^T B_{hyb}^{-1} (x - x_b) + \left(y - H_{(x)}\right)^T R^{-1} \left(y - H_{(x)}\right) \tag{1}$$

where $x$ is the analysis result; $x_b$ is the background field; $B_{hyb}$ is the hybrid background error covariance; $y$ is the GIIRS observation; $R$ is the observational error covariance; and $H$ is the observation operator, which converts the $x$ to the observation space. In this study, we choose RTTOV (Radiative Transfer for TOVS) fast radiative transfer model [29] version 13.1 as the observation operator to provide the DA system with simulated radiance and Jacobian ($K$) in channel space. Compared to the former version, RTTOV 13.1 allows the maximum surface skin temperature value over land to be 1250 K; this change can potentially reduce the radiance simulation error over wildfire, biomass burning, and high-fossil-fuel-consuming enterprises. Since the impact from other trace gases are removed in the channel selection, we use the "rtcoef_fy4_1_giirs_o3co2.H5" coefficient (available at https://nwp-saf.eumetsat.int/downloads/rtcoef_rttov13 (accessed on 31 December 2022)), which only considers the trace gas absorption impact from $O_3$ and $CO_2$, to generate the simulated GIIRS radiance and Jacobian. At each iteration ($n$), the first-order approximation can simplify the calculation of $H_{(x)}$ to:

$$H_{(x_n)} = H_{(x_{n-1})} + K_{(x_{n-1})} (x_n - x_{n-1}) \tag{2}$$

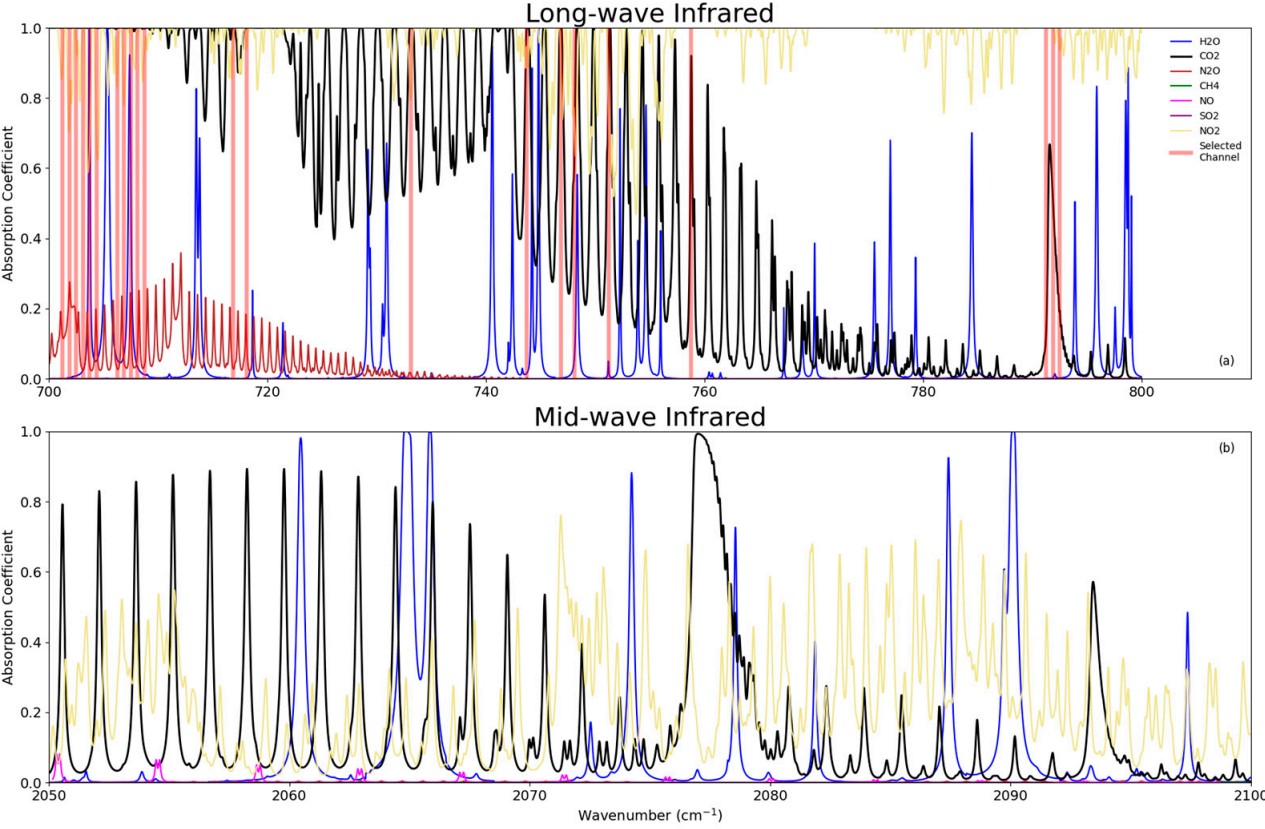

**Figure 1.** Gaseous absorption coefficient for $H_2O$ (blue), $CO_2$ (black), $N_2O$ (dark red), $CH_4$ (green), NO (pink), $SO_2$ (purple) and $NO_2$ (yellow) over GIIRS long-wave (**a**) and mid-wave (**b**) IR channels. Absorption data come from HITRAN2020 Dataset (available at https://hitran.org/ (accessed on 31 December 2022)).

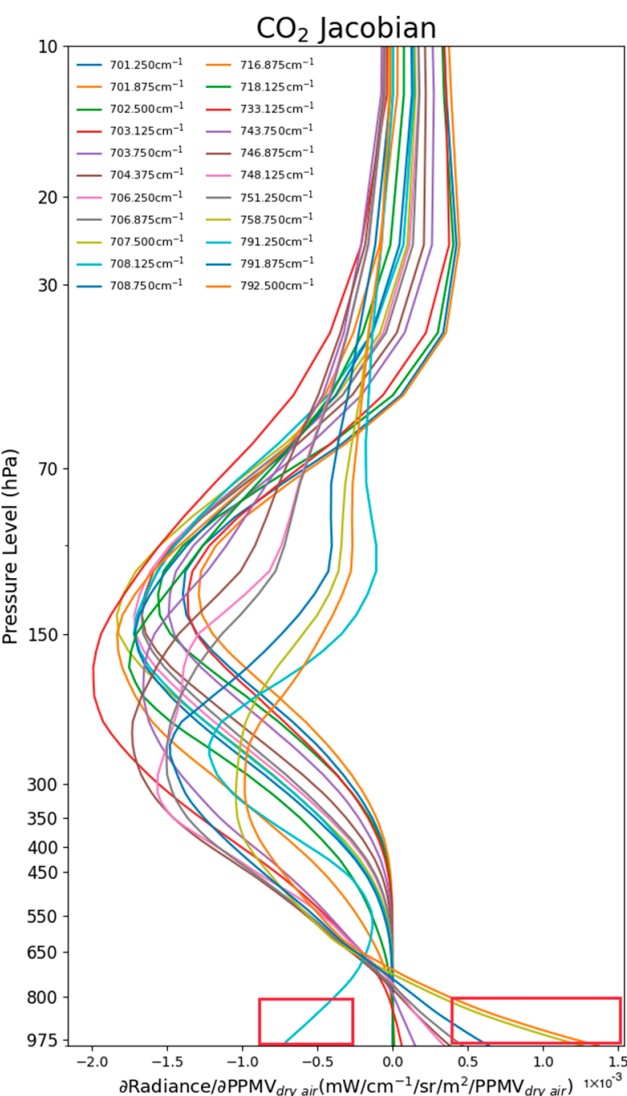

**Figure 2.** $CO_2$ Jacobian (weighting function) of the 22 selected GIIRS spectral channels.

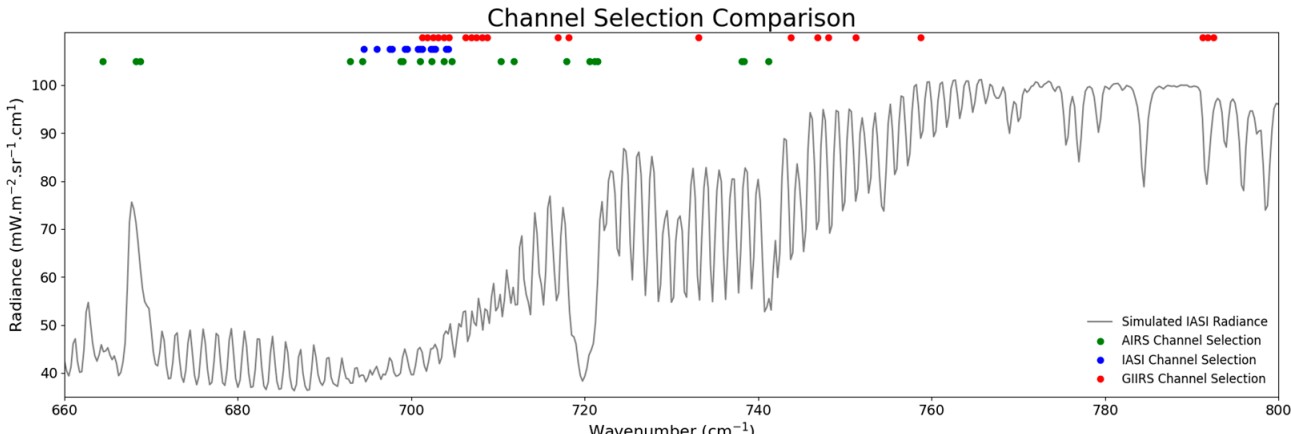

**Figure 3.** Channel selection scheme comparison between AIRS (green), IASI (blue) and GIIRS (red). The grey line is the mean radiance from simulated IASI observation.

In this study, the static background error covariance matrix ($B_{static}$) consists of three variables: surface skin temperature, air temperature, and $CO_2$ concentration (Figure 4a) is calculated from a three-month-long GEOS-5 forecast dataset via the NMC (National

Meteorological Center, now National Centers for Environmental Prediction) method [30]. The ensemble background error covariance ($B_{ensemble}$), which only consists of surface skin temperature and air temperature, is calculated from NOAA (National Oceanic and Atmospheric) Global Ensemble Forecast System (GEFS, data available at https://noaa-gefs-pds.s3.amazonaws.com/index.html (accessed on 31 December 2022)) [31] 48 h lead-time forecast via the ensemble-mean method [32,33]. The $B_{ensemble}$ can reduce the $CO_2$ retrieval accuracy uncertainty by better representing the meteorological field's uncertainty caused by localized weather systems. The meteorological part of $B_{hyb}$ includes 80% $B_{static}$ and 20% $B_{ensemble}$, but the $CO_2$ part is identical to the $B_{static}$ due to the shortage of atmospheric $CO_2$ forecast ensembles. The observation error covariance matrix ($R$) (Figure 4b) takes advantage of the Hollingworth–Lönnberg method [34,35], which is widely accepted and used in meteorological and atmospheric composition of the DA system [36–40]. As the observations located from 701.250 $cm^{-1}$ to 704.375 $cm^{-1}$, from 706.250 $cm^{-1}$ to 708.750 $cm^{-1}$, and from 791.250 $cm^{-1}$ to 792.500 $cm^{-1}$ are adjacent spectral channels, which causes a high cross-channel correlation (Figure 4c), the DA system cannot merely use the diagonal values in the $R$ matrix to minimize the cost function; thus, the off-diagonal values must be included. The cut-off iteration step amount is set to 200, accordingly, because of the inclusion of R Matrix off-diagonal values.

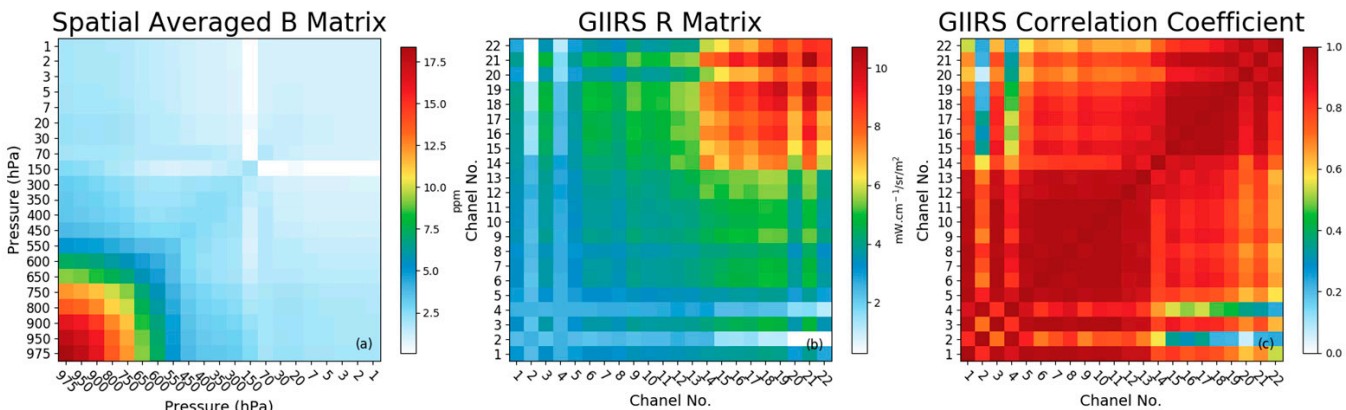

**Figure 4.** The spatial-averaged $CO_2$ B Matrix (**a**), the R Matrix (**b**) of the selected 22 GIIRS spectrums, and the spectral correlation coefficient matrix (**c**).

Before initiating the DA process, cloudy and high-noise-level pixel observations must be discarded. We quantify the cloud-covering possibility with Cloudy Pixel Percentage (*CPP*):

$$CPP = \frac{cloudy\ pixlel\ amount_{(searching\ radius,\ time\ window)}}{total\ pixel\ amount_{(searching\ radius,\ time\ window)}} \tag{3}$$

In the calculation, the cloudiness information comes from the FY-4A Advanced Geostationary Radiation Imager (AGRI, data available at https://satellite.nsmc.org.cn/portalsite/ (accessed on 31 December 2022)) China regional cloud mask product (4 km horizontal resolution, 7 min refresh rate) [41]. At each GIIRS observation location, the *total pixel amount* is the AGRI pixel amount within a 10 km spatial (14 min temporal) searching radius, and the *cloudy pixel amount* is the cloudy and possible cloudy AGRI pixel amount. Any observations with CPP higher than 0.1 cannot enter the Quality Control (QC) process. The QC method primarily relies on the Observation-minus-Background (OmB) estimation, which ensures that the observation-background departures are tolerant to the minimization algorithm and computational cost, especially when assimilating observations gathered from multiple instruments. Another approach to representing observation quality is using the Noise Equivalent spectral Radiation ($NE\Delta R$), which defines the amount of change in radiance required to produce a signal equivalent to the noise of the system. In this project, we adopt a $NE\Delta R$ -based QC method for two reasons: (1) smaller observation amount, where

the observations are gathered from 22 spectral channels via a single instrument; (2) data selection independence, where the OmB-based QC methods could bring about the over-similarity problem between observation and background and thus render the observation's impact on background modification. The QC method calculates the mean ($Mean_{NE\Delta R}$) and the standard deviation ($SD_{NE\Delta R}$) of each spectrum's $NE\Delta R$ from the clear-sky observations, then discards the entire pixel observation if it has more than 3 "bad" spectral observations, where bad means the absolute $NE\Delta R$ departure ($|NE\Delta R - Mean_{NE\Delta R}|$) is higher than 0.5 $SD_{NE\Delta R}$. In this study, the $NE\Delta R$-based QC method accepts 44.3% of the clear-sky observations, while the OmB-based method's data acceptance rate is 37.1%.

Based on the Emissions Database for Global Atmospheric Research (EDGAR) Green-House Gas (GHG) emission dataset [42], the experiment domain (Figure 5) in this study covers the primary GHG source regions in East Asia: China, Japan, and South Korea. The DA system assimilates GIIRS observation gathered within a 3 h time window ($\pm 1.5$ h relative to the analysis validation time) and generates analysis of $CO_2$ fields seven times per day (at 00, 03, 06, 09, 12, 15, 21 UTC). For performance comparison, the experimental analysis $CO_2$ concentration generated from the DA system and the background (GEOS-5 48-h lead-time forecast) are compared against GEOS-5 initial conditions (data available at https://portal.nccs.nasa.gov/datashare/gmao/geos-fp/das/ (accessed on 31 December 2022)) and OCO-2/-3 retrieval products (data available at https://disc.gsfc.nasa.gov/datasets/ (accessed on 31 December 2022)). Mean Bias (MB) and Standard Deviation (SD) are chosen to indicate the performance difference in the evaluation.

$$MB = \frac{\sum_{i=0}^{n} Bias}{n} \tag{4}$$

$$SD = \sqrt{\frac{\sum_{i=0}^{n}(Bias - MB)^2}{n}} \tag{5}$$

$$Bias = Estimation_{(Analysis \vee GEOS-5\ forecast)} - Truth_{(GEOS-5\ analysis \vee OCO-2/-3\ observation)} \tag{6}$$

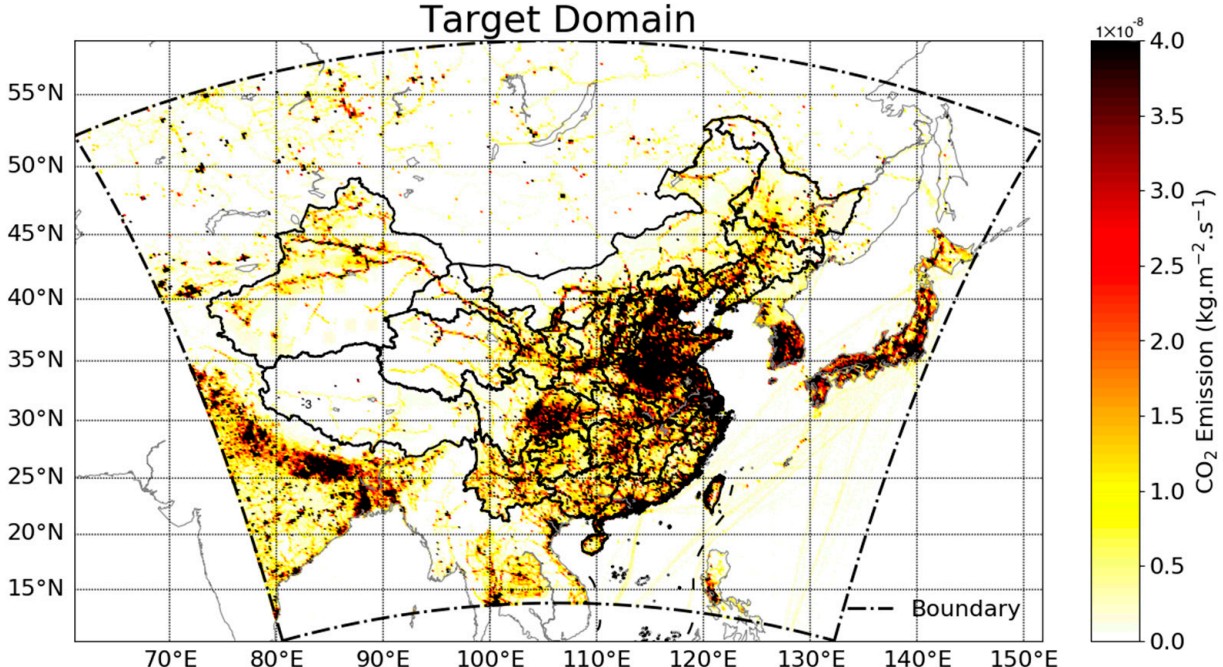

**Figure 5.** Domain spatial coverage and the grid-point GHG emission (data from EDGARv6.0, available at http://jeodpp.jrc.ec.europa.eu/ftp/jrc-opendata/EDGAR/datasets/v60_GHG/ (accessed on 31 December 2022)).

## 3. Results

### 3.1. Validation against GEOS-5 Initial Condition

By comparing the mean bias profile from the experimental analysis and GEOS-5 48 h lead-time forecast product (Figure 6a), it is shown that the GIIRS observation can increase the first guess's accuracy. Regarding the mean bias, the first-guess field from the GEOS-5 48 h lead-time forecast has a 0.27 ppm $CO_2$ concentration over-estimation at 975 hPa. This error is brought down by 0.04 ppm after assimilating the GIIRS observation. Furthermore, the low troposphere (975–800 hPa) benefits from the assimilation of the GIIRS observation since the experimental analysis's mean bias is smaller than the mean bias of the first guess. The impact of the GIIRS observation tends to be neutral over the mid-troposphere (700–500 hPa), as neither accuracy improvement nor downgrade can be detected in the experimental analysis product. This phenomenon could result from the shortage of mid-troposphere $CO_2$-sensitive spectral channels in the GIIRS observation (Figure 2). Unlike the observation shortage in the mid-troposphere, every selected spectral channel has one of its Jacobian peaks above 450 hPa. However, due to the relatively low $CO_2$ concentration in the upper troposphere, the added value from the GIIRS observation assimilation is smaller than that provided for the low troposphere. If the mean bias deduction is expressed as a percentage (Figure 6c), it shows that the GIIRS observation contributes more information to the upper-level $CO_2$ mean bias deduction, where the highest deduction ratio is almost 30%, whereas for the low troposphere, the highest deduction ratio is 27%. Rather than a positive contribution, the GIIRS observation increases the mid-troposphere mean bias by 5% (maximum). Despite the mid-troposphere performance drawback, the GIIRS observation still declares its potential in decreasing the first guess's $CO_2$ concentration bias on a broad scale. Other information concluded from the standard deviation profiles (Figure 6b,d) indicates that the GIIRS observation can diminish the random error in the background profiles.

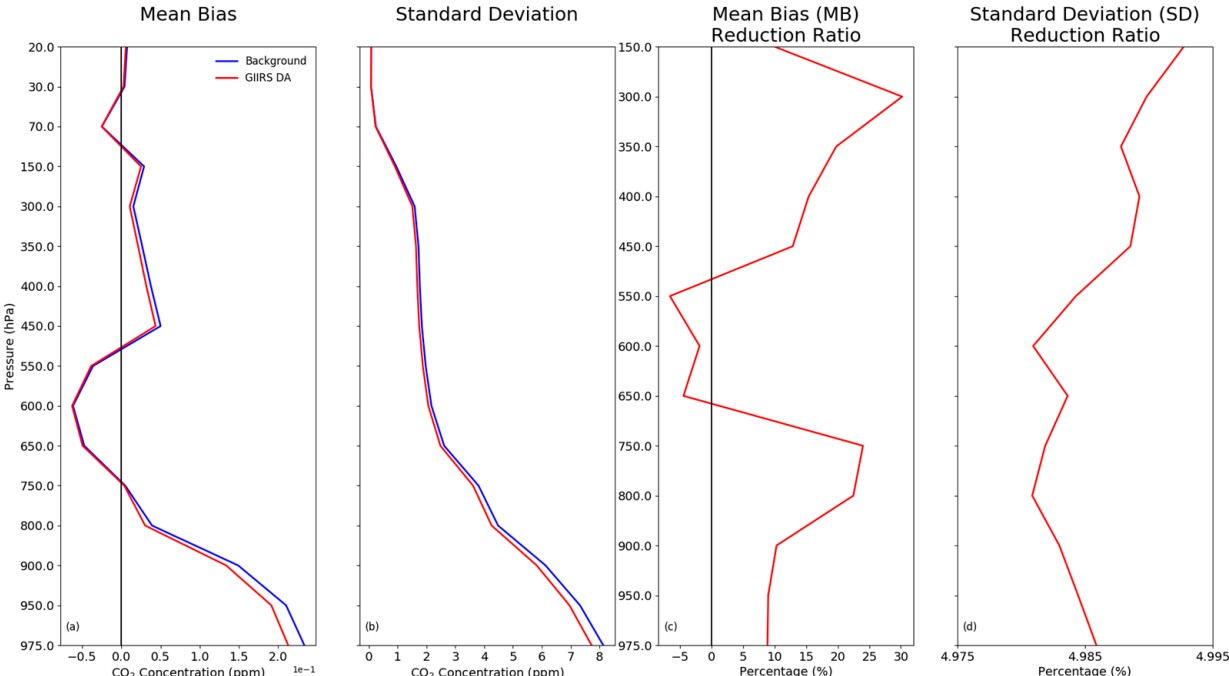

**Figure 6.** Mean bias profile (**a**) and standard deviation profile (**b**) before (blue) and after (red) assimilating GIIRS sounding observation for the domain shown in Figure 4 and from 1 August 2022 to 1 September 2022. The reduction ratio of mean bias and standard deviation are shown in (**c**,**d**). Solid black lines in subplot (**a**,**c**) are the ideal mean bias profile.

Since the low troposphere (975–800 hPa) $CO_2$ concentration is highly related to human activities, it is useful to show how the GIIRS observation can detect the diurnal cycle. The diurnal change in mean bias (Figure 7a) indicates a higher value during midnight and twilight, but this anomaly drops below 0.16 ppm during the daytime. This situation happens not only in the first guess but also the experimental analysis product, but the DA system brings the analysis field's mean bias down by 0.016 ppm (20:00 BJT/12:00 UTC) to 0.035 ppm (08:00 BJT/00:00 UTC) by assimilating the GIIRS observation. At 23:00 BJT/15:00 UTC, the GIIRS observation increases the mean bias, which can be attributed to the instrument's insatiability before the GIIRS scheduled a 2-hour-long offline time centered at 02:00 BJT/18:00 UTC. The standard deviation diurnal cycle (Figure 7b) shares the same pattern with the mean bias (Figure 7a), in which the GIIRS observation contributes a 0.12 ppm (20:00 BJT/12:00 UTC) to 0.34 ppm (08:00 BJT/00:00 UTC) standard deviation deduction upon the first guess. Unlike the mean bias result at 23:00 BJT/15:00 UTC, the experimental analysis's standard deviation is smaller than the first guess, but with a marginal magnitude.

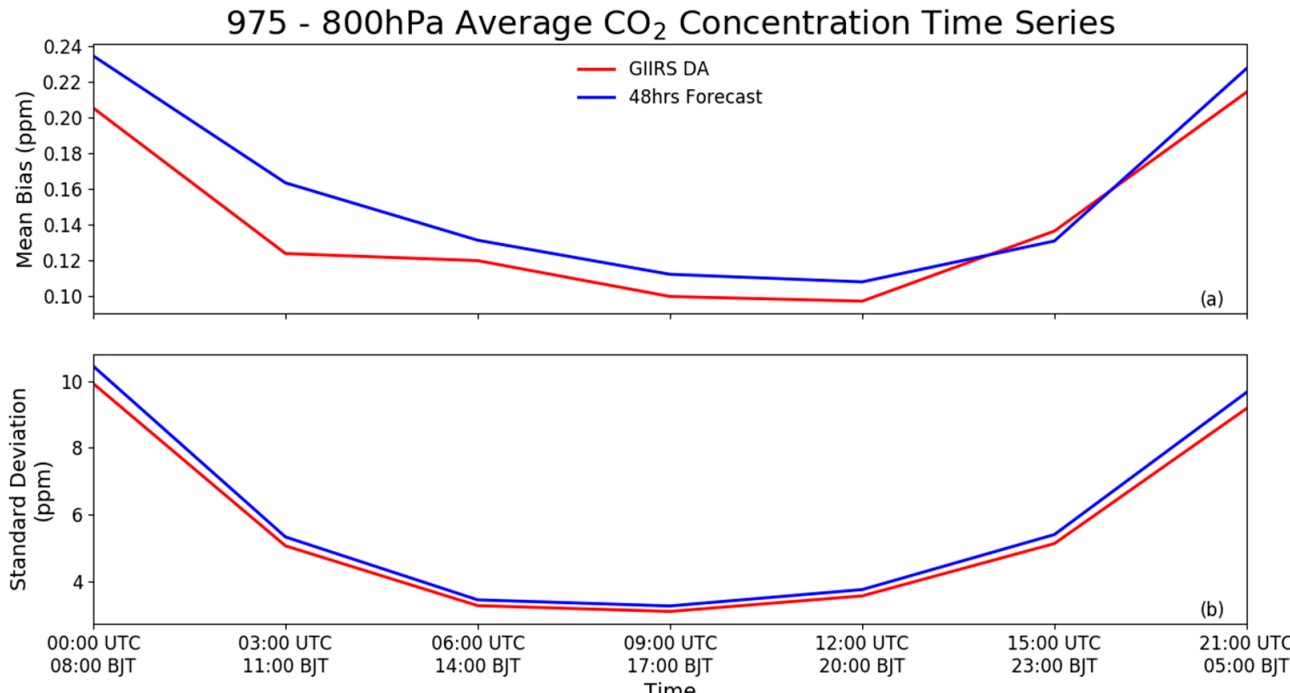

**Figure 7.** Diurnal cycle of mean bias (**a**) and standard deviation (**b**) before (blue) and after (red) GIIRS sounding assimilation, for the domain shown in Figure 4 and from 1 August 2022 to 1 September 2022.

The low-troposphere $CO_2$ added value (experimental analysis minus first guess) spatial distribution (Figure 8) depicts the experimental analysis product lowers (increases) the $CO_2$ concentration in the first guess during local daytime (nighttime). This spatial distribution shows where the GIIRS observation modifies the first guess to minimize the bias in the low-troposphere $CO_2$ concentration field: in the daytime, the primary $CO_2$ concentration reductions being located in west and southeast China (Figure 8a), while at night, south China produces more $CO_2$ emissions (Figure 8b).

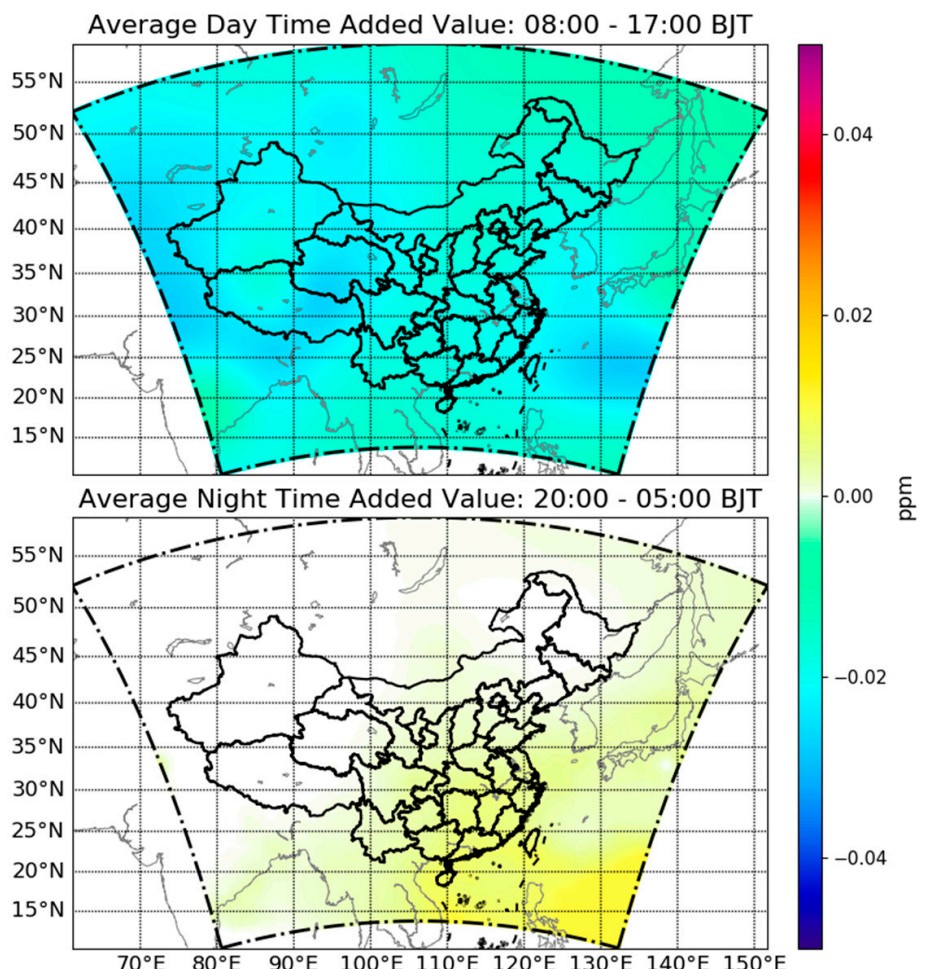

**Figure 8.** Spatial distribution of daytime and nighttime mean bias added value.

As can be seen from Figure 9a, assimilation of GIIRS observations lowers the daytime low-troposphere standard deviation spatial distribution over most regions in China, except the west and central areas, where complex terrain is located. Comparing the random error reduction with the GHG emission (Figure 5), it can be concluded that the places with high standard deviation reduction coincide with high GHG-emission regions. The phenomenon indicates that assimilating the GIIRS observation can reduce the $CO_2$ concentration random error where the GHG emission level is high. The nighttime standard deviation reduction spatial distribution (Figure 9b) differs from the daytime one: the random error increases after GIIRS assimilation in north and northeast China; and the tropical northwest Pacific Ocean (lower right corner) has the highest standard deviation increment above that for other regions. The human activity uncertainties (primarily industrial and high-energy-absorption activities) can be one of the causes of the nighttime random-error increase over north and northeast China since these regions have the most extended heavy industry history in China. For the random error increase over the tropical northwest Pacific, the coefficient for the GIIRS instrument could be the cause: the default GIIRS coefficient in RTTOV is calculated from a global training set, but according to the results from Di et al. [43], a coefficient calculated from a localized training set can improve fast radiative transfer models' simulation accuracy since the geostationary hyperspectral infrared sounder is supposed to be more localized than other polar-orbiting ones.

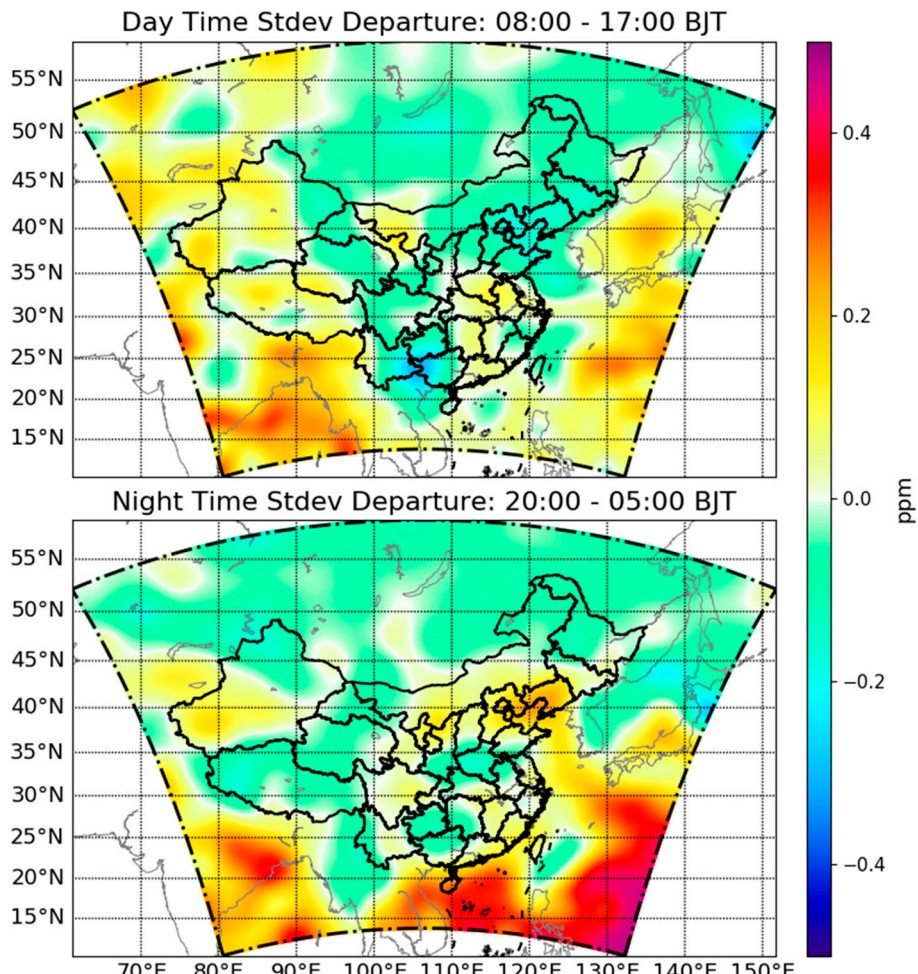

**Figure 9.** Spatial distribution of daytime and nighttime standard deviation departure.

### 3.2. Validation against OCO-2/3 Observation

The former section demonstrates the GIIRS observation capability of correcting the $CO_2$ concentration's systematic error (mean bias) and random bias (standard deviation) in the first guess. However, the analysis field still needs further validation against observations, e.g., retrieval products generated from polar-orbiting platforms or in situ observations. However, the comparison against in situ observation is highly difficult to conduct due to the observation shortage in targeted area. In this section, we compare the mean bias and standard deviation from the experimental analysis and the GEOS-5 analysis, and the GEOS-5 48-hour lead-time forecasts are used to demonstrate the GIIRS observation's impact on $CO_2$ concentration correction and to calculate the mean bias and standard deviation of the differences with the level-2 $CO_2$ retrieval products from OCO-2 [44] and OCO-3 [45] being treated as an unbiased observation (i.e., "Truth"). Before calculating the bias, the OCO-2/-3 products are mapped relative to the estimation (GIIRS DA analysis, GEOS-5 analysis, or the first guess) location where only the spatiotemporally nearest retrieval is used. Figure 10 shows the geographical locations of OCO-2/-3 retrievals used in this evaluation. As can be seen, the OCO-2 retrieval dataset provides observations over high-latitude regions (52°N and north), with more homogeneous spatial distribution than OCO-3, which covers the experiment domain's northwest. This phenomenon could induce disagreement between evaluation results against OCO-2 and OCO-3 products.

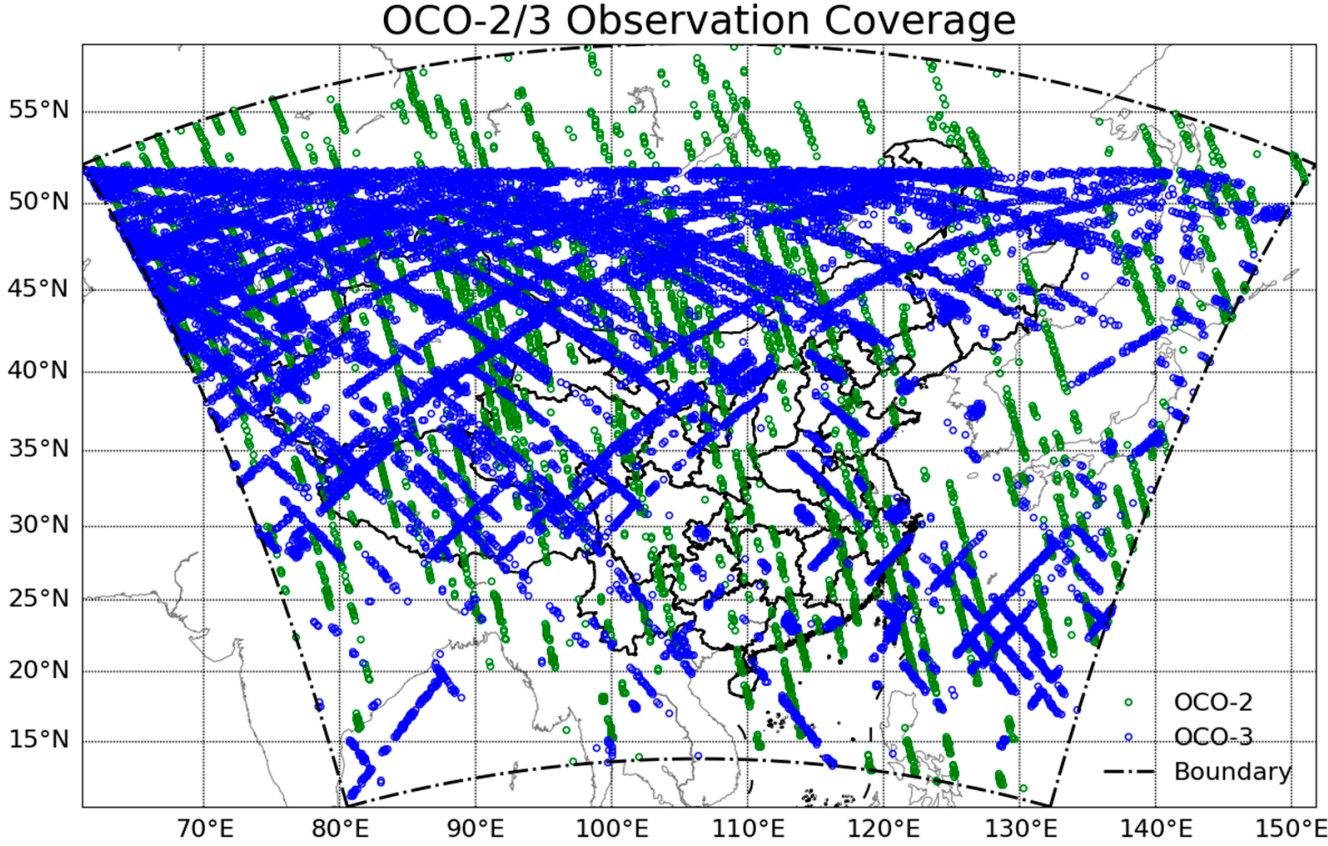

**Figure 10.** OCO-2 (green) and OCO-3 (blue) retrieval profile location used in the evaluation.

From the mean bias profile (Figure 11a,c), all three estimations (experimental analysis, GEOS-5 analysis, and the first guess) underestimate the low-troposphere $CO_2$ concentration but produce overestimation over upper and mid-troposphere. Meanwhile, the magnitude of the mean bias is more extensive than that in the former section, which could be related to spatial difference: the OCO-2/-3 instruments have a 1.29 km × 2.25 km horizontal resolution, but a pixel in the GEOS-5 product/GIIRS observation is (25 km × 25 km)/(16 km × 16 km). Nevertheless, differences between the experimental analysis (red line), GEOS-5 analysis (black line), and the first guess (blue line) still reveals that the GIIRS analysis result has equivalent performance to the GEOS-5 analysis, while the first guess's departure from the GEOS-5 analysis result is not neglectable (Figure 11a,c). The performance difference in random bias cancellation is more noticeable than the mean bias: in the OCO-2 comparison (Figure 11b,d), the GIIRS observation can cut down the $CO_2$ concentration's random error in the first guess over the low troposphere.

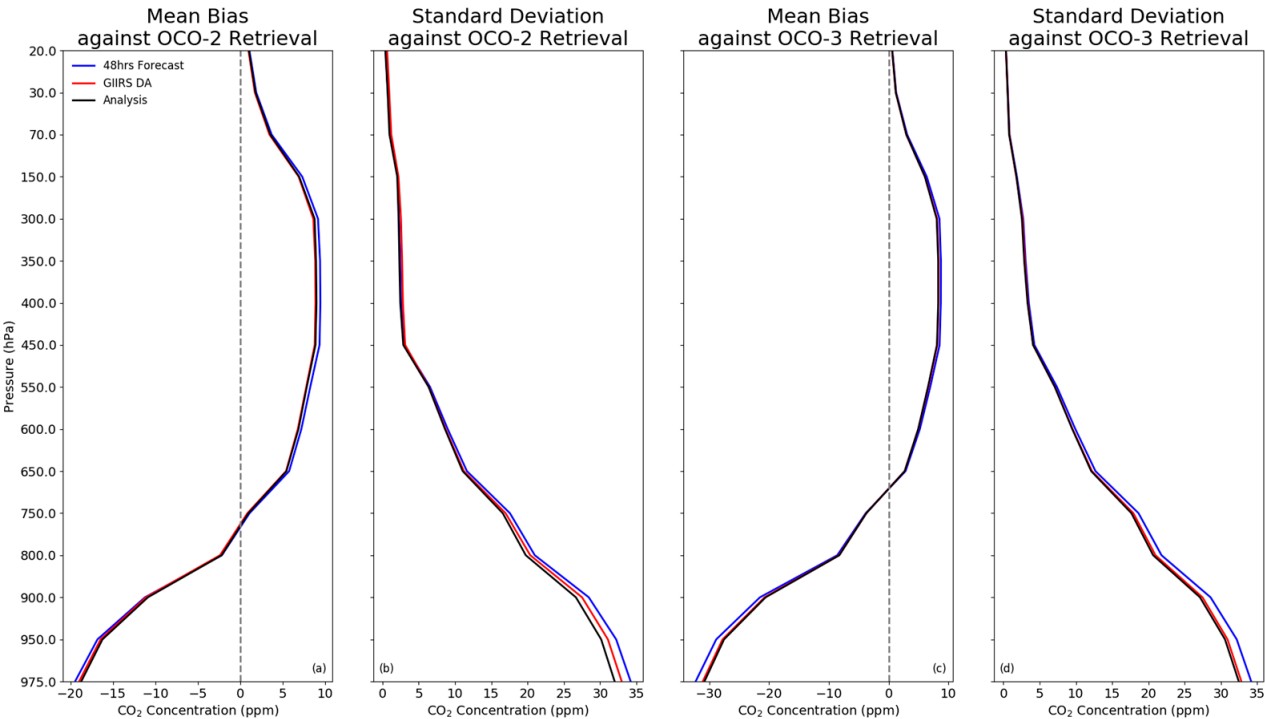

**Figure 11.** Mean bias profile of GIIRS assimilation (red), first guess (blue) and GEOS-5 analysis (or GEOS-5 initial condition) (black) with OCO-2 (**a**) or OCO-3 (**c**) as the truth; the standard deviation profiles using OCO-2 (OCO-3) as the truth are shown in (**b**,**d**). Dotted lines in subplot **a** and **c** are the ideal mean bias profile.

## 4. Discussion

As mentioned in the beginning, geostationary instruments can diminish the revisiting time latency and provide observations at the same footprint within every spatial scan time; these attributes give geostationary hyperspectral infrared sounders multiple advantages over polar-orbiting instruments [46–48]. Meanwhile, the horizontal resolution of the GIIRS instrument impedes its contribution in increasing the horizontal resolution of $CO_2$ monitoring: the instrument on FY-4A has a 16 km horizontal resolution at the nadir point, and its successor, GIIRS, on FY-4B's nadir point horizontal resolution is 12 km, which is roughly equivalent to GOSAT but far behind the OCO-2/-3 instrument. However, this situation will change after Meteosat Third-Generation InfraRed Sounder (MTG-IRS) and Geostationary Extended Observation Sounder (GXS) become operational. In the future, the likelihood of creating a global-scale high-spatial–temporal-resolution $CO_2$ monitoring network is probable with a global constellation of meteorological geostationary hyperspectral infrared sounder instruments. This satellite constellation can provide useful insights into global greenhouse gas emission reduction. For retrospective $CO_2$ analysis datasets, meteorological geostationary hyperspectral infrared sounders can undoubtedly improve their temporal resolution by combining the new observation with the existing datasets from OCO-2 and GOSAT. Since the FY-4B GIIRS instrument's $NE\Delta R$ is much smaller than that of FY-4A (Figure 12), the quality of this $CO_2$ retrieval dataset can be improved by adding its observation into the system.

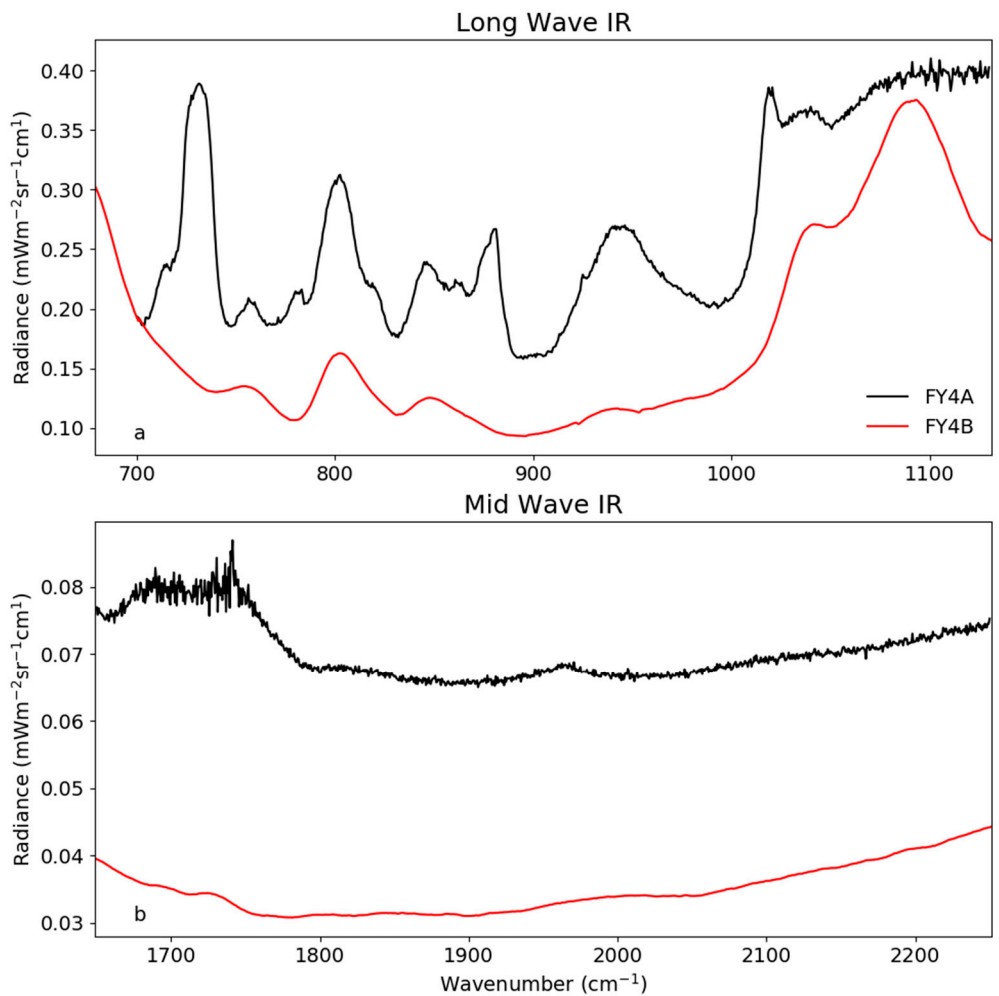

**Figure 12.** GIIRS noise equivalent differential radiance from FY-4A (black) and FY-4B (red) in long-wave infrared (**a**) channel and mid-wave infrared channel (**b**).

## 5. Conclusions

In this research, we conducted a prototypical study to demonstrate the geostationary hyperspectral infrared sounder's potential for monitoring $CO_2$ concentration using a one-month-long GIIRS observation. The results indicate that the meteorological geostationary infrared sounding observations can provide $CO_2$ concentration information with reasonable accuracy. With high revisiting time frequency, GIIRS is capable of revealing the diurnal variation of $CO_2$ emission. The additional spectrum, 751.250 $cm^{-1}$, 758.750 $cm^{-1}$, 791.250 $cm^{-1}$, 791.875 $cm^{-1}$ and 792.500 $cm^{-1}$, can reduce the uncertainties embedded in the GEOS-5 product over the low troposphere. Despite the findings listed above, more research and investigations are needed to solve the issues such as the parallax correction for the geostationary hyperspectral infrared observations and how to combine the advantages from geostationary and low-earth-orbiting hyperspectral infrared sounding observation together.

**Author Contributions:** Conceptualization, Q.Z. and W.S.S.; methodology, Q.Z. and W.S.S.; software, Q.Z.; validation, Q.Z.; resources, Q.Z.; writing—original draft preparation, Q.Z.; writing—review and editing, W.S.S., M.S. All authors have read and agreed to the published version of the manuscript.

**Funding:** The research is funded by Special Science and Technology Innovation Program for Carbon Peak and Carbon Neutralization of Jiangsu Province (Grant Number BE2022612).

**Institutional Review Board Statement:** Not applicable.

**Informed Consent Statement:** Not applicable.

**Data Availability Statement:** The online access URLs are provided at the first location where the data are mentioned in the article.

**Acknowledgments:** We thank Filomena Romano and Elisabetta Ricciardelli for their voluntary academic editing services. We would also like to show our gratitude to all "anonymous" reviewers for their insights and comments on an earlier version of the manuscript, although any errors are our own and should not tarnish the reputations of these esteemed persons.

**Conflicts of Interest:** The authors declare no conflict of interest.

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
