# Peer review of "The Potential of Monitoring Carbon Dioxide Emission in a Geostationary View with the GIIRS Meteorological Hyperspectral Infrared Sounder"

_remotesensing, doi:10.3390/rs15040886_

Round 1
Reviewer 1 Report
In a whole, it gives an interesting research showing the geostationary hyperspectral infrared sounder’s potential for monitoring CO2 concentration using GIIRS observation. However, the method and explanation on results is not clear. Please clarify these procedures and make this paper read-friendly for potential reader. Here it is the specific comments or suggestions:
Line 66. a ‘spectrum’ resampling
Figure 1: please add the physical variation of y-axis. It is more likely the transmittance rather than the absorption coefficient.
Line 69: please clarify the procedure for selecting the channels. And it is also not sure that how the CO2 Jacobians to be calculated.
Line 78: a typo ‘CO2‘. Please double check the similar typo otherwhere.
Table 1. Maybe a figure plotting all channel selected for three instruments overlay a spectram is read-friendly than a table for potential reader.
Line 123: please check ?h?? , if you mean B ?????? ?
Line 143: please rewrite this sentence, it is quite confusing.
Line 158: not sure how authors calculate the the standard deviation (????∆?) for each spectrum or channel at each pixel, it seems only has one value of NEdR at this moment. Or you mean whole GIIRS observation within one time window.
Line 170: quite confusing here, because you mention before that there are no CO2 forecast GEOS-5 48-hours lead-time forecast. And what is the GEOS-5 initial condition? So what is exactly the background CO2 profile for GSI assimilation?
Line 184: same question as before. Why the GEOS-5 initial condition could be using as a reference.
Figure 10. Similar question on legend and data. So if the 48hr forecast that you refer is first guass, so it is so-called GEOS-5 initial condition? What is the GOES-analysis? So it is the analysis from DA model without GIIRS included? If it is not, please clarify and discuss if it is necessary to include a control run (DA but no GIIRS included).
Reviewer 2 Report
Dear authors,
Thank you for your exploration of how we can better understand GIIRS measuring carbon dioxide concentration. Here are some suggestions that should improve the quality of your manuscript.
1. In Abstract, lines 19-20, acronym GEOS-5 and OCO-2/-3 should be explained there, in Abstract, since it was the first time mentioning that in this manuscript. 2. In Introduction, line 38, the acronym explanation should be in Abstract, not here. 3. In Introduction, line 53, GIIRS was explained already in Abstract. 4. In Materials and Methods, line 114, it should be consistant through whole manuscript the first acronym or the first explanation as the authors commenced in Abstract. 5. Results should be consisting only results, the Discussion chapter should contain most of the comments after the results. 6. I suggest to authors to change the title of the chapter Summary into Conclusions, since conclusions include more scientific approach. 7. In Conclusion, the authors should use the word findings and write what were the findings in this manuscript. 8. In Conclusions, line 331, the authors should never say what the authors' future plan should be. All sciencists in this field should be addressed instead. 9. In Conclusions, the authors should advice future researchers of what could be done in the future for more improvement.Author Response
Please see the attachment.

Reviewer 3 Report
This paper present a prototypical study to demonstrate the geostationary hyperspectral infrared sounder’s potential for monitoring CO 2 concentration using one-month-long GIIRS observation. The results indicate that the meteorological geostationary infrared sounding observations can provide CO 2 concentration information with reasonable accuracy. The article is well written, but some more information and a few more references would have been good.
Line 86: The table shows many empty lines
Line 106: Please give some more information about the RTTOV setting.
Line 143: too many empty spaces
Line 144: please a reference for the AGRI cloud mask and also a brief description
Line 169: The GIIRS scanning frequency ten times per day (at 0, 2, 4, 6, 8, 10, 12, 14, 20, and 22 h UTC) but for your assimilation you have chosen the following seven times per day00, 03, 06, 09, 12, 15, 21 UTC, if the purpose of the article is the high temporal resolution monitoring of CO2 why not use all 10 acquisitions.
